# The Therapeutic Potential of Kiwi Extract as a Source of Cysteine Protease Inhibitors on DNCB-Induced Atopic Dermatitis in Mice and Human Keratinocyte HaCaT Cells

**DOI:** 10.3390/ijms26041534

**Published:** 2025-02-12

**Authors:** Hye Ryeon Yang, Most Nusrat Zahan, Du Hyeon Hwang, Ramachandran Loganathan Mohan Prakash, Deva Asirvatham Ravi, Il-Hwa Hong, Woo Hyun Kim, Jong-Hyun Kim, Euikyung Kim, Changkeun Kang

**Affiliations:** 1Department of Basic Veterinary Medicine, College of Veterinary Medicine, Gyeongsang National University, Jinju 52828, Republic of Korea; tkwk565@naver.com (H.R.Y.); zahan.nusrat.gnu@gmail.com (M.N.Z.); pooh9922@hanmail.net (D.H.H.); mohan_22@gnu.ac.kr (R.L.M.P.); devabiochem@gnu.ac.kr (D.A.R.); ihhong@gnu.ac.kr (I.-H.H.); woohyun.kim@gnu.ac.kr (W.H.K.); jkim@gnu.ac.kr (J.-H.K.); ekim@gnu.ac.kr (E.K.); 2Institute of Animal Medicine, Gyeongsang National University, Jinju 52828, Republic of Korea

**Keywords:** kiwifruit, cysteine protease inhibitor, DNCB, atopic dermatitis, skin lesion, human keratinocyte, TNF-α

## Abstract

The discovery of effective cysteine protease inhibitors with crude protein kiwi extracts (CPKEs) has created novel challenges and prospects for pharmaceutical development. Despite extensive research on CPKEs, limited research has been conducted on treating atopic dermatitis (AD). Therefore, the objective of this work was to investigate the anti-inflammatory effects of CPKEs on TNF-α activation in a HaCaT cell model and in a DNCB (1-chloro-2, 4-dinitrochlorobenzene)-induced atopic dermatitis animal model. The molecular weight of the CPKE was determined using SDS-PAGE under non-reducing (17 kDa and 22 kDa) and reducing conditions (25 kDa, 22 kDa, and 15 kDa), whereas gelatin zymography was performed to examine the CPKE’s inhibitory impact on cysteine protease (actinidin and papain) activity. Moreover, the CPKE remains stable at 60 °C, with pH levels varying from 4 to 11, as determined by the azocasein assay. CPKE treatment decreased the phosphorylation of mitogen-activated protein kinase (MAPK) and Akt, along with the activation of nuclear factor-kappa B (NF-κB)-p65 in tumor necrosis factor-α (TNF-α)-stimulated HaCaT cells. Five-week-old BALB/c mice were treated with DNCB to act as an AD-like animal model. The topical application of CPKE to DNCB-treated mice for three weeks substantially decreased clinical dermatitis severity and epidermal thickness and reduced eosinophil infiltration and mast cells into ear and skin tissues. These findings imply that CPKE derived from kiwifruit might be a promising therapy option for inflammatory skin diseases such as AD.

## 1. Introduction

In recent years, natural products and plant-derived compounds have concentrated significant attention on their potential as therapeutic agents in managing inflammatory skin disorders [1]. Atopic dermatitis (AD) is a chronic inflammatory skin disease characterized by excoriation, edema, and itching and is often colloquially referred to as eczema [2]. It affects a substantial portion of the global population and poses significant challenges to both patients and healthcare providers due to its complex etiology and variable clinical manifestations [3]. The development of AD is accomplished and comprises genetic influences, impaired epidermal barrier function, disrupted microbiome balance, immunological irregularities, and inflammation [4]. Research conducted on the quality of life among children and young adults has consistently found that AD has adverse effects on various aspects of their physical, emotional, and social well-being [5]. The prevailing therapeutic approach for visible skin lesions is through anti-inflammatory treatments, typically using steroids [6]. Topical corticosteroids and calcineurin inhibitors, such as tacrolimus and pimecrolimus, are frequently used together [7].

Topical treatments for AD encompass corticosteroids, which effectively reduce inflammation and pruritus, and topical calcineurin inhibitors, such as tacrolimus and pimecrolimus, which provide long-term management with a reduced risk of side effects compared to corticosteroids [8]. Emollients and moisturizers, particularly those containing ceramides, hyaluronic acid, or glycerin, play a critical role in restoring the skin barrier and alleviating dryness [9]. Topical antihistamines can be utilized to mitigate pruritus, while topical antimicrobials, including mupirocin and antiseptic agents, are indicated to address secondary bacterial infections [10]. Additionally, pine tar, a traditional therapy, remains effective in reducing scaling, inflammation, and pruritus in chronic cases [11,12], while newer biologic therapies, such as crisaborole (a PDE4 inhibitor), show promise in managing mild-to-moderate AD by reducing inflammation and improving skin integrity [13].

While these topical treatments can effectively relieve symptoms of AD, mitigate inflammation, and deter flare-ups, they come with potential side effects when used over an extended period [13]. Lately, there has been a rising interest in exploring alternative treatment options for AD, particularly the natural bioactive compounds derived from plant extracts [14]. Despite extensive research, the exact underlying mechanisms of AD remain incompletely understood, hampering the development of targeted therapeutic interventions [15]. Several research have implicated the dysregulation of protease activities in AD’s pathophysiology [16]. Variations in the genes producing proteases and protease inhibitors or abnormalities in protease activity have been identified as contributing factors to the development of AD [16]. Exogenous cysteine proteases have been verified to possess the ability to directly disrupt the tight junctions in epithelial tissues [17].

Defective filaggrin, resulting from *FLG* gene mutations, compromises the skin’s barrier integrity, leading to elevated trans epidermal water loss and increased susceptibility to allergen and microbe invasion [18]. Additionally, variations in immune-related genes such as IL-4, IL-13, and TNF-α skew the immune response toward a Th2-dominant profile, exacerbating inflammation [19]. This compromised barrier function, influenced by both genetic predisposition and environmental triggers, plays a pivotal role in the pathogenesis of atopic dermatitis.

Environmental triggers in atopic dermatitis include allergens, microbes, and irritants, all of which can worsen symptoms [19]. These external factors, such as pollen, pet dander, dust mites, and certain chemicals, can provoke immune responses that intensify inflammation, leading to flare-ups of the condition [20]. Additionally, microbial infections, particularly with *Staphylococcus aureus*, are common in AD patients and contribute to the severity of symptoms by further disrupting the skin barrier and promoting inflammation [21].

Kiwifruit, native to Asia, is known for its rich content of bioactive molecules and gained widespread popularity owing to its exceptional sensory attributes and nutritional qualities [22]. Kiwifruit, commonly known as "kiwi," is the edible berry of plants in the Actinidiaceae family, and the two main species used for commercial production are *Actinidia deliciosa* (fuzzy kiwifruit) and *Actinidia chinensis* (golden kiwifruit) [23]. *A*. *deliciosa* is a rich source of bioactive compounds, including cysteine protease inhibitors that have been shown to possess various health-promoting properties [24]. Also renowned for its rich nutritional profile and bioactive compounds, including vitamin C, polyphenols, flavonoids, and carotenoids, which contribute to its potent antioxidant and health-promoting properties [25,26].

Cysteine protease inhibitors derived from *A*. *deliciosa* have demonstrated potential in modulating protease activity and associated inflammatory responses [27]. Kiwifruit extract, particularly its bioactive compounds, has demonstrated significant anti-inflammatory effects by modulating cytokine activity, which plays a crucial role in immune responses and inflammatory disorders [28]. Studies have shown that kiwifruit extract inhibits the production of pro-inflammatory cytokines such as TNF-α, IL-6, and IL-1β, which are key mediators of inflammation [29]. Additionally, kiwifruit extract has been found to enhance the secretion of anti-inflammatory cytokines like IL-10, thereby promoting a balanced immune response [30,31]. These inhibitors have the capacity to modulate the activity of cysteine proteases and, by extension, mitigate the inflammatory cascade associated with AD [17]. CPKE has the capacity to regulate the activity of cysteine protease and thereby plays an important role as a therapeutic intervention [32]. Additionally, the potential role of CPKE is to suppress inflammatory pathways by targeting proteases involved in the activation of pro-inflammatory mediators such as cytokines, chemokines, and inflammatory enzymes [33,34].

Our present study investigates the therapeutic potential of kiwifruit extract as a natural source of cysteine protease inhibitors in the context of AD. To elucidate the mechanisms underlying the observed effects, we examine in vitro studies to show the results of an anti-inflammatory efficacy on keratinocytes stimulated by TNF-α. On top of that, we conduct in vivo studies to examine the impact of CPKE in an AD mouse model, specifically focusing on its influence on the immune modulating effect in this model and comparing other therapeutic agents like dexamethasone.

Additionally, the efficacy of CPKE was assessed through clinical skin severity scores and histological changes in an in vivo model, while its molecular effects were evaluated by analyzing the phosphorylation of MAPK, Akt, and NF-κB-p65 in TNF-α-induced HaCaT cells during in vitro studies. Our aim is to contribute to building up a significant amount of evidence for using natural compounds from kiwifruit as a means of ameliorating AD symptoms, potentially offering a safe and effective alternative or adjunct therapy to conventional treatments.

## 2. Results

### 2.1. Purification of Cysteine Protease Inhibitor from Kiwifruit

The cysteine protease inhibitor from kiwifruit was successfully purified. This involved extraction, precipitation with ammonium sulfate, dissolution in sodium phosphate buffer, dialysis, acetone treatment, centrifugation, rotary vacuum evaporation, and lyophilization. The purified protein’s high concentration and purity were confirmed using analytical techniques, ensuring its quality and integrity. The yield of the purified cysteine protease inhibitor from kiwifruit (100 g) was determined to be 2.46 g or 2.46%. 

### 2.2. SDS-PAGE Profile of CPKE

Model of SDS-PAGE gel electrophoresis acquired in both reducing and non-reducing status. Under non-reducing conditions for SDS-PAGE, the CPKE resulted in two bands, and the molecular weight was found to be 17 kDa and 22 kDa (Figure 1A). But, reducing SDS-PAGE of the same sample exhibited several bands of molecular weights 25 kDa, 22 kDa, and 15 kDa, respectively (Figure 1B). It indicates the presence of the enzyme as a dimer (connected by disulfide bridges) in its natural form for CPKE.

### 2.3. Determination of Optimal Temperature and pH Stability for CPKE

For the determination of optimal pH for CPKE activity, the enzyme’s performance was evaluated across varying buffer systems with different pH values (as explained in the section on materials and methods). All pH levels were ideal for CPKE activity (Figure 2A). Additionally, CPKE displayed maximal enzymatic activity in the pH range of 4–11. Experiments were carried out to evaluate the effectiveness of CPKE activity at different temperature settings to identify the ideal temperature. As shown in Figure 2B, the enzyme was preincubated for 60 min at temperatures ranging from 4 to 80 °C to examine the thermal stability of CPKE. At 60 °C, CPKE showed stability; however, at 80 °C, its activity decreased. Preincubating at a higher temperature (80 °C) for 60 min led to a slight decrease in activity. These findings indicate that CPKE activity rises with higher pH and temperature levels, reaching its peak at pH 11.0 and 60 °C, which encourages future research into the enzyme’s stability of CPKE.

### 2.4. Inhibition Effect of CPKE Against Cysteine Proteases

We investigated the inhibition activity of CPKE against cysteine protease using different ratios. Gelatin zymography was used to evaluate CPKE’s capacity to suppress cysteine protease activity. Gelatin served as the substrate in the proteolytic zymography assay. Two cysteine proteases, actinidin and papain, were utilized to illustrate CPKE’s suppression of cysteine protease activity. Actinidin is from kiwifruit-derived cysteine protease, and papain is like actinidin, so we used those cysteine proteases in this experiment. Varied ratios of pre-incubated mixtures demonstrated decreasing proteolytic activity as the proportion of CPKE’s to actinidin or papain increased, indicating CPKE’s ability to suppress cysteine protease activity effectively. As the ratio of CPKE rose, a dose-dependent decline in cysteine protease inhibitory activity was noted, as shown in Figure 3. These observations indicate that CPKE effectively inhibits the activity of cysteine proteases.

### 2.5. The Assessment of Cell Viability Assay for the Treatment of HaCaT Cells

An MTT test was used to assess the optimal CPKE treatment concentration for HaCaT cells. After 24 h at a concentration of 50 μg/mL, the CPKE showed no signs of cytotoxicity. But, after 24 h, the cell viability of those treated with a 1000 μg/mL CPKE concentration dropped by around 67%, as seen in Figure 4A. Additionally, we evaluated the effect of CPKE with and without TNF-α. Even when subjected to 10 ng/mL TNF-α activation in HaCaT cells, CPKE showed no cytotoxicity up to a dose of 100 μg/mL (Figure 4B). Concentrations of 50 and 100 μg/mL of CPKE combined with 10 ng/mL of TNF-α were selected for additional research based on the observed cytotoxicity profile. Specifically, in our experimental system, these amounts showed no cytotoxicity. Consequently, these doses were considered appropriate for additional research.

### 2.6. TNF-Induced Alterations in the MAPK Family and NF- κB p65 Pathway of Proteins in HaCaT Cells Are Modulated by CPKE Treatment

We first examined whether CPKE suppresses the molecular basis of anti-inflammatory activity in TNF-α stimulated HaCaT cells and requires the activation of the Akt and mitogen-activated protein kinase (MAPK) signaling pathways. Before subjecting HaCaT cells to TNF-α stimulation for 30 min, a pretreatment of CPKE (at levels between 50 and 100 μg/mL) was administered for 1 h. Through Western blot analysis, we investigated the phosphorylation levels of ERK 1/2, p38, and Akt. As shown in Figure 5A, CPKE treatment resulted in a dose-dependent decrease in p38, ERK 1/2, and Akt phosphorylation in cells stimulated with TNF-α. Also, we examined the phosphorylation of NF-κB p65 activation in HaCaT cells stimulated with TNF-α. The transcriptionally active component p65 within the NF-κB complex undergoes phosphorylation at the serine 536 residue to regulate the transcriptional activation of NF-κB. CPKE treatment exhibited a dose-dependent prevention of the translocation of NF-κB p65, as depicted in Figure 5B. Moreover, as illustrated by the representative blot image, it is evident that CPKE, when administered without TNF-alpha stimulation, induces a concentration-dependent increase in the phosphorylation of both AKT and ERK. However, it should be noted that without TNF-alpha stimulation, it showed opposite effects. The observed effect was more pronounced with higher CPKE concentrations, demonstrating that CPKE can modulate these signaling pathways even in the absence of TNF-alpha stimulation. These results demonstrate that CPKE has an anti-inflammatory impact on the buildup of allergy modulators via blocking Akt MAPK and suppressing NF-κB p65 activation.

### 2.7. CPKE Improved the Clinical Symptoms of AD-like Skin Lesions in DNCB-Induced Mice

DNCB was administered to BALB/c mice to induce skin lesions resembling AD. Hence, the mice were exposed to both CPKE and dexamethasone (positive control), as illustrated in Figure 6. CPKE was conducted with a low dose of 2.5 mg/mL and a high dose of 5 mg/mL. This experimental setup aimed to evaluate the effectiveness of CPKE in mitigating these skin lesions. The application of DNCB results in substantial inflammation, as was expected. AD impacts the immune system because it typically appears as a systemic immunological response. Therefore, to ascertain if topical CPKE treatment has an anti-AD effect in mice, we measured the weight of the lymph node and spleen throughout the course of the previous week. In comparison to the sham group, the control group caused an increase in the weight of the lymph nodes and spleen in the mice. The CPKE treatment groups suppressed lymph node weight in a dose-dependent manner (Figure 7A). Additionally, following CPKE treatment, spleen weight was somewhat reduced (Figure 7B). These findings thus demonstrate that CPKE can provide anti-AD benefits when applied topically.

### 2.8. The Curative Effect of CPKE on AD-Like Skin Lesions in Mice

AD led to an increase in dermal lymphocyte infiltration and skin thickness. H&E and toluidine blue were used to stain the skin and the ear to determine whether CPKE can treat cutaneous etiology similar to AD. H&E staining showed that the epidermal and dermal tissues in the control group both markedly grew in thickness, whereas the epidermal tissue in the CPKE treatment group dramatically decreased in thickness compared to the control group. Furthermore, toluidine blue staining showed a considerable quantity of mast cells in the dermal region, while the CPKE treatment group’s mast cell count decreased in a dose-dependent manner (Figure 8). These results therefore advance our knowledge of the etiology of the mouse AD-like skin disease.

## 3. Discussion

The present study investigates the therapeutic potential of kiwifruit extract, specifically its cysteine protease inhibitor with crude protein kiwi extract (CPKE), in mitigating symptoms and inflammation associated with DNCB-induced AD. Over the past few years, there has been extensive research into the natural properties and healthcare facilities of kiwifruit, exploring its diverse range of actions, including gastrointestinal disorders, cancer, constipation, cardiovascular disease, and diabetes [24,35,36,37]. Due to polyphenolic composition and biological activities, kiwifruit has antioxidant, antimicrobial, and antiproliferative properties [23,38]. Also, several studies conducted on kiwifruit-induced allergy, which is prevalent in both children and adults, are chiefly attributed to actinidin, a papain-like cysteine protease, constituting up to 50% of soluble proteins [24,25]. Due to polyphenolic composition and biological activities, kiwifruit has antioxidant, antimicrobial, and antiproliferative properties [23].

An interesting fact is that our study uncovers a fascinating aspect: the extraction of a cysteine protease inhibitor from kiwifruit. This inhibitor demonstrates the capacity to mitigate allergic reactions, presenting a novel therapeutic candidate specifically for atopic dermatitis. The major findings of our study are that CPKE discovered from kiwifruit extract acts as a potent cysteine protease inhibitor, with implications for inhibiting other proteolytic enzymes. In addition, CPKE can inhibit the activation of Akt, MAPKs, and NF-κB activation in HaCaT cells stimulated by TNF-α, which can also reduce the development of DNCB-induced AD lesions in BALB/c mice. While CPKE was shown to attenuate MAPK and NF-κB signaling pathways in the presence of TNF-alpha, we recognize that, in the absence of TNF-alpha, CPKE can stimulate p-AKT and p-ERK in a concentration-dependent manner. This finding suggests that, under certain conditions (e.g., without the inflammatory trigger TNF-alpha), CPKE may exhibit a pro-inflammatory effect. The activation of these pathways is typically associated with cell survival, proliferation, and inflammation, and they are important for driving certain cellular responses during injury and inflammation. On top of that, future research should focus on further understanding how CPKE interacts with other signaling pathways and identifying the conditions under which it switches between pro- and anti-inflammatory actions. Our previous study demonstrated that BALB/c mice induced with DNCB exhibited a pathogenesis resembling human AD, characterized by erythema, abnormalities in skin thickness, and immunologic dysregulation [14]. To evaluate the effect of CPKE on the degree of inflammatory skin and histological alterations, we used the DNCB-induced BALB/c mice model. During the experiment, we observed notable inflammatory responses after repeated topical administration of DNCB on dorsal skin and ear tissue.

Current therapies for AD, such as topical corticosteroids and calcineurin inhibitors, are effective in reducing inflammation and suppressing immune responses but are often associated with limitations, including adverse effects like skin atrophy, tachyphylaxis, and systemic immunosuppression risks with prolonged use [39]. Calcineurin inhibitors, while beneficial in reducing cytokine production, are expensive and less accessible for long-term use in some populations [40]. In comparison, CPKEs present a novel mechanism of action by specifically targeting the dysregulated protease activity implicated in epidermal barrier dysfunction [41]. By restoring protease-antiprotease balance, CPKEs offer a dual advantage: preserving skin integrity and reducing inflammation without direct immunosuppression [33]. This mechanism positions CPKEs as complementary or alternative agents to existing therapies, particularly for patients seeking treatments with fewer side effects or for use in maintenance therapy. The ability of CPKEs to attenuate MAPK and NF-κB signaling cascades not only highlights their role in reducing inflammatory cytokine production but also suggests potential benefits in preventing disease exacerbation and chronicity. Additionally, the natural origin of the kiwifruit-derived CPKE adds an appealing dimension, particularly for individuals seeking plant-based or non-steroidal options. Future studies could explore the integration of CPKE-based formulations with existing treatments, such as combining them with emollients or as adjuncts to corticosteroids, to assess synergistic effects and optimize outcomes. SDS-PAGE analysis of the CPKE revealed that, under non-reducing conditions, the protein existed as a dimer with molecular weights of 17 kDa and 22 kDa. Under reducing conditions, the CPKE dissociated into multiple bands, including 25 kDa, 22 kDa, and 15 kDa, indicating the presence of monomers. These findings suggest that the CPKE naturally forms a dimer stabilized by disulfide bonds, which may be crucial for its stability and activity of CPKE.

In addition, the optimal temperature and pH conditions for assessing the enzymatic functions, biological activity, and related properties of CPKE were determined. The CPKE showed stability up to 60 °C, but its activity slightly decreased when exposed to 80 °C. For the determination of optimal pH for CPKE activity, the enzyme’s performance was evaluated across varying buffer systems with different pH values (as specified in the section on materials and procedures). All pH levels were ideal for CPKE activity. Furthermore, the pH rose in each group in a dose-dependent pattern. CPKE also showed peak enzymatic activity across a broad range of pH (4–11). In addition, to check the inhibition activity of CPKEs, we used cysteine proteases, namely actinidin and papain. This experiment was evaluated by using gelatin zymography. The CPKE effectively suppresses the activity of cysteine proteases as its concentration increases, indicating its ability to inhibit cysteine protease activity.

Keratinocytes, for instance, HaCaT cells, are crucial in skin disease and serve as standard models for testing anti-inflammatory properties in vitro [42]. Stimulation of keratinocytes by TNF-α activates MAPK pathways, including ERK1/2 and p38, crucial for cell survival and inflammatory responses [43]. Numerous physiological processes, including cell survival, proliferation, differentiation, metabolism, apoptosis, stress response, and inflammation, are influenced by the MAPK pathways [44]. Several extracellular stimuli, for instance, cytokines, growth factors, and environmental stressors, are responsible for the activation of MAPK signaling [45]. Activation of MAPKs facilitates the phosphorylation of subsequent targets, for instance, transcription factors and pro-inflammatory mediators, thereby intensifying inflammatory signals, with dysregulation of the MAPK pathway induced by various inflammatory diseases [46]. Likewise, the PI3K (phosphoinositide 3-kinase)/Akt pathway, commonly referred to as the Akt pathway, is essential for controlling inflammation [47]. 

Regarding the inflammatory response, Akt activation facilitates the recruitment of immune cells to the site of inflammation and increases the production of pro-inflammatory cytokines. [48]. Additionally, Akt signaling enhances the inflammatory response by interacting with other inflammatory pathways like NF-κB [48]. Dysregulated Akt signaling is responsible for causing the pathogenesis of inflammatory diseases [49]. Furthermore, by interacting with other immune signaling pathways, such as NF-κB, Akt signaling amplifies the inflammatory response [50]. In reaction to inflammatory stimuli such as cytokines, microbial products, and oxidative stress, NF-κB is activated and translocated to the nucleus [51]. Next, it triggers the transcription of genes that promote inflammation, including adhesion molecules, cytokines, and chemokines. As importantly, the genesis, propagation, and resolution of inflammation are facilitated by the MAPK, Akt, and NF-κB pathways [52]. Targeting these pathways can be a potent therapeutic approach for developing novel therapeutic strategies for inflammatory diseases.

In HaCaT cells, CPKE treatment prevents TNF-α-induced phosphorylation of Akt, ERK, and p38, indicating that it may have anti-inflammatory properties. Furthermore, we analyzed NF-κB p65 activation, which is crucial for inflammatory gene regulation. CPKE dose-dependently inhibited NF-κB p65 translocation, indicating its anti-inflammatory potential. 

We next assessed CPKE’s impact on the extent of skin damage and associated histopathological alterations in DNCB-induced mice. CPKE in 3 weeks of treatment significantly improved skin symptoms, reduced dorsal and ear skin thickness, and decreased lymph node and spleen weight. In comparison to the sham group, the control group caused an increase in the lymph nodes and spleen weight in the mice. The CPKE treatment groups suppressed lymph node weight in a dose-dependent way. Additionally, following CPKE treatment, spleen weight was somewhat reduced. These findings thus demonstrate that CPKE can provide anti-AD benefits when applied topically. Remarkably, the CPKE treatment group demonstrated a marked decrease in case of tissue thickness compared to the control, suggesting a potential mitigating effect on AD-induced skin changes. Moreover, toluidine blue staining unveiled a decrease in the dermal region, and mast cell count was reduced in the control group, indicative of inflammatory responses associated with AD. These results not only confirm the CPKE’s therapeutic efficacy for CPKEs on skin conditions like AD but also suggest a dose-dependent relationship, reinforcing the potential efficacy of CPKE in attenuating cutaneous manifestations skin to AD.

Moreover, CPKE application attenuated inflammatory eosinophil and mast cell infiltration in the ears, stated its therapeutic potential in AD-like lesions, and its influence on immune cell dynamics. 

Since excessive protease activity has been reported to be another characteristic abnormality that affects the epidermal barrier in inflammatory skin disease, several studies also reported that a proteolytic cascade leads to the desquamation process, and its progression might result in the initiating event in skin inflammation [53]. Moreover, inflammatory-related cytokines are activated by a proteolytic process, which can commit the inflammatory response [54,55]. Therefore, protease inhibitors have emerged as a promising therapeutic strategy for managing skin diseases, particularly those characterized by inflammation and barrier dysfunction, such as AD. These inhibitors target proteolytic enzymes involved in skin barrier degradation, immune response dysregulation, and inflammation. For instance, serine protease inhibitors have been shown to restore epidermal barrier integrity and reduce inflammatory cytokine levels [56]. The efficacy of protease inhibitors in preclinical and clinical studies underscores their potential for targeted intervention in chronic inflammatory skin conditions. 

Results from both in vitro and in vivo studies support the effectiveness of CPKE as a treatment for AD-like skin lesions in BALB/c mice given DNCB. This study uniquely explores the anti-inflammatory and therapeutic potential of a natural cysteine protease inhibitor (CPKE) derived from kiwifruit, an underutilized resource in dermatological research. By employing both in vitro (HaCaT cells stimulated with TNF-α) and in vivo (DNCB-induced AD model in BALB/c mice) approaches, the findings provide robust evidence of CPKE’s efficacy across complementary experimental systems. Notably, this dual-modality investigation allowed us to elucidate the molecular mechanisms underlying CPKE’s action, including modulation of the MAPK and NF-κB signaling pathways, which are central to AD pathogenesis.

## 4. Materials and Methods

### 4.1. Chemicals and Reagents

Dimethyl Sulfoxide (DMSO) and 3-(4,5-dimethylthiazol-2-yl)-2,5-diphenyltetrazolium bromide (MTT) were sourced from Sigma-Aldrich Inc. (St. Louis, MO, USA). Bovine Serum Albumin (BSA) and Dulbecco’s Modified Eagle’s Medium (DMEM) were acquired from Gibco-BRL (Grand Island, NY, USA). Antibodies targeting phospho-Akt (Ser473), MAPK sampler kit, phospho-NF-κB p65, and β-actin were procured from Cell Signaling Technology (Beverly, MA, USA). TNF-α was obtained from R&D Systems (Tokyo, Japan). All other reagents employed were of the highest possible purity.

### 4.2. Purification of Cysteine Protease Inhibitor

A cysteine protease inhibitor from kiwifruit was purified using a slightly modified method of Siddiqui (2016) [57]. The kiwifruit (100 g) was ground into 300 mL of extraction buffer (50 mM sodium phosphate buffer pH 7.5, 3 mM EDTA, and 0.15 M sodium chloride) in a grinder and extracted overnight at room temperature. The extract was centrifuged by cooling at 8000 rpm for 5 min. Pellets were thrown away. Overnight, 70% ammonium sulfate saturation was used to separate the supernatant. Centrifugation at 8000 rpm for 45 min at 4 °C was used to collect the precipitate, which was then dissolved in at least 50 mM sodium phosphate buffer (pH 7.5). To eliminate ammonium sulfate, the precipitated protein was thoroughly dialyzed against sodium phosphate buffer (pH 7.5) at 4 °C. After treating the diluted sample with cold acetone in a 1:1 ratio, it was centrifuged for 20 min at 8000 rpm. A rotating vacuum evaporator (Tokyo Rikakikai Co., Ltd., Tokyo, Japan) was used to evaporate the supernatant. Lyophilization under vacuum to dryness was the last stage.

### 4.3. SDS-PAGE Profile of CPKE

Electrophoresis was carried out using a 4% stacking gel and a 12% separating gel in accordance with the Laemmli technique [19]. The sample buffer (4% SDS, 125 mM Tris-HCl, pH 6.8, 20% glycerol, 0.01% bromophenol blue) was used to prepare the CPKE sample, which was then kept at −20 °C until it was needed. Using a Tris-glycine running buffer, CPKE was electrophoresed for 90 min at 100 V at steady voltages. For molecular weight determination, the Precision Plus ProteinTM Standards, 10–250 kDa (Bio-rad, Hercules, CA, USA), were run in tandem with the CPKE. After electrophoresis, 0.125% Coomassie blue in 40% methanol and 10% acetic acid were used to stain the separated protein bands. 

### 4.4. Proteolytic Activity of CPKE

Proteolytic activity was assessed using the modified Segers et al. protocol [20]. Amounts of 100 µL of CPKE, 150 µL of reaction sodium bicarbonate buffer (0.5% solution, pH 8.3) (Buffer A), and 250 µL of 2.5% azocasein (*w*/*v*) dissolved in Buffer A were all included in the reaction mixture. The assays were carried out at 37 °C and stopped after 30 min by adding 400 µL of 10% (*w*/*v*) trichloroacetic acid. The reaction mixture was centrifuged at 12,000 rpm for 20 min to remove the precipitated protein. By adding 300 µL of 500 mM NaOH and measuring absorbance at 440 (Power Wave TMXS, Bio-Tec Instruments, Inc., Winooski, VT, USA), the 500 µL supernatant was neutralized.

### 4.5. The Impact of Temperature and pH on the Stability and Activity of CPKE

The ideal temperature and pH for CPKE were found using an enzyme experiment that was performed by following Aissaoui, N. et al. with slight modification [58]. In order to select the optimal temperature, the enzymatic examination was performed at different temperatures (4–80 °C). An aliquot (100 mg) of CPKE was dissolved in 1 mL solvent of distilled water (DW). The CPKE was preincubated for 60 min at temperatures ranging from 4 to 80 °C to assess thermal stability. Assaying in buffers with several pH values at 4 °C (pH 4; acetate buffer, pH 7; phosphate buffer, pH 10.0 and 11.0; glycine–NaOH buffer) allowed for the determination of the ideal pH. The CPKE was maintained at 4 °C for 30 min in several buffers with pH values ranging from 4.0 to 11.0 to test for pH stability. As previously explained, residual proteolytic activity was calculated and expressed as a percentage of the original activity, which was assumed to be 100.

### 4.6. Examination of Cysteine Protease Inhibition Activity Using Zymography

The inhibitory activity of CPKE was determined using a slightly modified method of Kunitz (1947) [59]. The protein concentrations of the CPKE, ficin, and papain were assessed using the Bradford method (Bio-Rad, Hercules, CA, USA), and the CPKE dose was calculated based on this protein concentration. Gelatin was used as a substrate for the gelatin zymography assay. In order to create the corresponding zymography gels, gelatin (2 mg/mL) dissolved in 20 mM sodium phosphate buffer (pH 7.4) was copolymerized with 12% polyacrylamide. The indicated concentration of actinidin and papain (100 µg/mL) was incubated with 5:5, 6:4, 7:3, 8:2, 9:1, 10:0, and a 0:10 ratio of the purified CPKE (100 µg/mL) for 30 min at 37 °C. After being prepared in a non-reducing sample buffer, the incubation samples for analysis were run on gels at 100 V and 4 °C. Following electrophoresis, the gel was cleaned twice for 30 min in 2.5% Triton X-100 to remove SDS. The gel was then stained with 0.125% Coomassie blue after being incubated for 16 h at 37 °C with 20 mM Tris (pH 7.4), 0.5 mM calcium chloride, and 200 mM sodium chloride. Regions of proteolytic activity are indicated as clear zones in the gel.

### 4.7. Cell Culture and Cell Viability

Human keratinocyte (HaCaT) cells were obtained from the American Type Culture Collection (ATCC). Cell viability was assessed using the MTT assay. Briefly, HaCaT cells were cultured in DMEM supplemented with 10% FBS, 100 μg/mL penicillin–streptomycin, and maintained at 37 °C in a humidified incubator with 5% CO_2_. After that, HaCaT cells were cultivated in 24-well plates at a density of 4 × 10^4^ cells/well, and they were left to adhere to the growth DMEM mix for the entire night. After gently washing the cells with fresh culture medium, they were subjected to various concentrations of CPKE and then incubated for 24 h. Subsequently, 5 mg/mL of MTT solutions were introduced into each well, followed by an additional 3-h incubation at 37 °C. Ultimately, DMSO was introduced to facilitate the solubilization of the formazan salt, and the quantity of generated formazan was assessed through optical density measurements. (OD) at 540 nm using a GENios® microplate spectrophotometer (Power Wave TMXS, BioTek Instruments, Inc., Winooski, VT, USA).

### 4.8. Western Blot Analysis

In six-well plates, cells were cultivated at a density of 5 × 10^4^ cells per well, and they were left to incubate in full DMEM for 24 h. Following the adaptation process, cells were exposed to CPKE for a duration of 1 h. Subsequently, they were stimulated with TNF-α (10 ng/mL) for a period of 2 h using a serum-free culture medium. The CPKE was maintained in the culture medium throughout the experiment to evaluate its continuous effect on the cells during the inflammatory stimulation with TNF-alpha. This approach allowed us to assess the sustained activity of the CPKE in modulating the inflammatory response over the course of the treatment. The treated cells were collected by gently scraping them with 300 μL of RIPA buffer (Trans Lab, Daejeon, Republic of Korea) supplemented with a protease inhibitor after being washed with cold PBS. The lysates were separated on a 10% SDS–polyacrylamide gel and subsequently transferred to PVDF membranes (Bio-Rad, Hercules, CA, USA). Specific primary antibodies were applied to the Western blot and allowed to probe overnight at 4 °C. A secondary antibody conjugated with horseradish peroxidase (Bethyl Laboratories Inc., Montgomery, AL, USA) was applied to the membranes for one hour at room temperature following the incubation of the main antibody. Chemi-Doc XRS (Bio-Rad, Hercules, CA, USA) was used to evaluate the blot signals once discovered using an enhanced chemiluminescence technique (ECL, Amersham Biosciences, Buckinghamshire, UK). Using a mage Lab software version 4.1 (Bio-Rad, Hercules, CA, USA), a densitometry study was carried out.

### 4.9. Experimental Animals

We kept the five-week-old female BALB/c mice at the laboratory animal research facility of Gyeongsang National University after purchasing them from Samtako Inc. (Osan, Republic of Korea) for maintenance and care. Under standard conditions, the mice were housed in cages with unlimited access to food and water and were exposed to a 12-h light/dark cycle. The ambient temperature and humidity were maintained at 23 ± 2 °C and 35–60%, respectively. The research involving animals in this study was granted approval by the Institutional Animal Care and Use Committee of Gyeongsang National University. The assigned protocol number for the animal study is GNU-181122-M0060, approved on 22 November 2018.

### 4.10. Experimental Study with CPKE for AD

In total, 5 groups (*n* = 7) of 35 female mice were randomly assigned to the following groups: group 1 was the vehicle-treated group (sham); group 2 was the vehicle with DNCB (control); groups 3 and 4 were the 2.5 mg/mL and 5 mg/mL CPKE with DNCB, respectively; and group 5 was the dexamethasone with DNCB group (positive control). The CPKE dose selection was based on the clinical pathology identified in our preliminary toxicity evaluation. Induction of atopic dermatitis in the mice was achieved through the administration of 1-chloro-2, 4-dinitrochlorobenzene (DNCB). On the first day, mice from every group had their dorsal skin hair cut off with an electric razor. During the first three days, mice in control, CPKE, and dexamethasone groups received 200 μL of a 0.5% DNCB solution once daily to the dorsal skin and ears. The solution was made by dissolving it in a 3:1 combination of acetone and olive oil. Only acetone and olive oil were used as a vehicle treatment for the sham group. CPKE was administered topically to the mice’s ears and back every day for three weeks after the initial sensitization treatment. Mice treated with dexamethasone as positive control received intraperitoneal injections of 1.5 mg/kg three times per week. On the 35th day, the animals were sacrificed. Figure 6 depicts the precise experimental schedule.

### 4.11. Measurement of Ear Thickness and Organ Weight

On the day of sacrifice, ear thickness was measured using a micrometer. The thickness was measured in micrometers, and the micrometer was placed near the tip of the ear, directly distal to the cartilaginous ridges. An electronic balance was used to determine the weights of the spleen and lymph nodes.

### 4.12. Histopathological Studies

After slicing the ear and skin lesions, the collected tissue slices were submerged for 24 h in 10% buffered neutral formalin. The tissue slices that had been fixed were enclosed in paraffin wax, subsequently cut into sections, subjected to deparaffinization, and then rehydrated, following established protocols. Sections, each measuring 5 μm in thickness, underwent staining procedures involving hematoxylin and eosin (H&E) as well as toluidine blue. The sections were then placed on glass slides and subjected to deparaffinization, involving xylene treatment to remove paraffin wax. Rehydration followed, achieved by passing the slides through decreasing concentrations of ethanol and eventually into water, restoring the tissue to a hydrophilic state for staining. To distinguish between various kinds of inflammatory cells, several stains were used. Using light microscopy, histological alterations were evaluated. An arbitrary scope was assigned to each tiny field seen at magnifications of 100 for the skin and 200 for the ears.

### 4.13. Statistical Analysis

The findings are presented as mean ± standard deviation (S.D.) and were derived from at least three repeated independent experiments. Using SPSS version 13, the *t*-test and one-way analysis of variance (ANOVA) was used to determine the statistical analysis, and then Dunnett’s post hoc test was performed. *p*-values below 0.05 were regarded as statistically significant.

## 5. Conclusions

Our findings emphasize the potential of the cysteine protease inhibitor from kiwifruit as a potential medicinal intervention for AD. The observed ameliorative effects in DNCB-induced mice and TNF-ɑ stimulated HaCaT cells, coupled with their impact on key inflammatory mediators, empower continued exploration of this natural compound as a viable candidate for managing AD treatment in clinical practice. The suppression of clinical severity in AD skin symptoms, along with a reduction in dermal lymphocyte infiltration and skin thickness, demonstrated the efficacy of CPKE in alleviating pathological changes associated with AD. We have found encouraging indications that the cysteine protease inhibitor has therapeutic promise for treating AD by combining in vitro and in vivo investigations. Further research and clinical trials, along with molecular mechanisms, are required to understand and facilitate the development of targeted interventions for AD.

One limitation of this study is the inconsistency in incubation times across various experimental treatments. While the pre-incubation period varied, this may have affected the comparability of the results, particularly in assessing the full extent of the therapeutic effects of the CPKE in atopic dermatitis models. Standardizing incubation times is crucial for ensuring reproducibility and minimizing variability in biological responses. In future experiments, we will extend the pre-incubation period to 24 h and ensure uniformity in both pre-incubation and post-treatment durations to enhance methodological consistency.

## Figures and Tables

**Figure 1 ijms-26-01534-f001:**
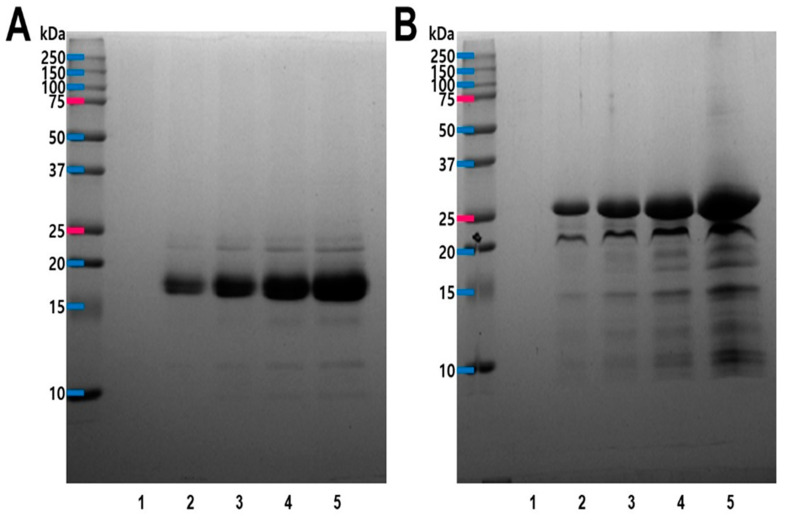
SDS-PAGE profile of CPKE. SDS electrophoresis was performed on CPKE in both (**A**) non-reducing and (**B**) reducing conditions. A 0.125% Coomassie blue stain was applied to the gels. The gel lanes were filled with varying doses of CPKE: 0 µg/mL in lane 1, 10 µg/mL in lane 2, 20 µg/mL in lane 3, 40 µg/mL in lane 4, and 80 µg/mL in lane 5.

**Figure 2 ijms-26-01534-f002:**
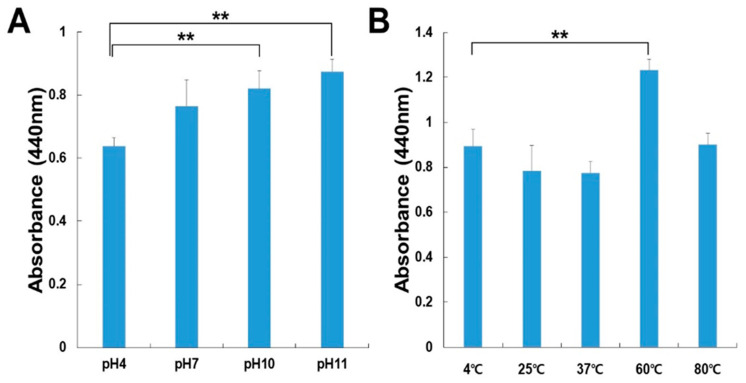
The impact of pH and temperature on the enzymatic activity of CPKE. Azocasein assays were used to assess the enzyme activity at 440 nm. An amount of 1 ml of solution (DW) was used to dissolve 100 mg of CPKE. (**A**) Incubation at 37 °C for 30 min across a pH range of 4, 7, 10, and 11 was used to measure the CPKE activity. (**B**) Following incubation at temperatures ranging from 4 °C to 70 °C, the CPKE activity was measured. The mean ± SD of three separate experiments is displayed. Statistically significant difference from the control group, ** *p* < 0.01, compared to each group using Dunnett’s test and ANOVA.

**Figure 3 ijms-26-01534-f003:**
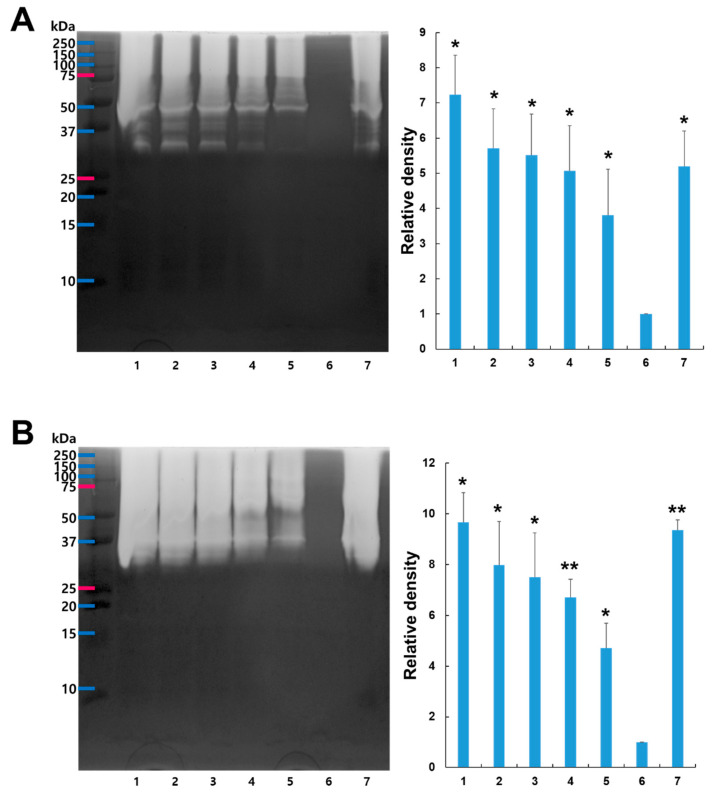
Inhibition effect of cysteine protease activity of CPKE. Gelatin zymography of pre-incubated mixtures at varied ratios was used to confirm the inhibition of CPKE cysteine protease activity. CPKE inhibits the activity of cysteine proteases, as demonstrated using (**A**) actinidin and (**B**) papain, one of the cysteine proteases. Lanes 1 through 5:5, 2 through 6:4, 3 through 7:3, 4 through 8:2, 5 through 9:1, 6 through 10:0, and 7 through 0:10 (ratio of CPKE to actinidin or papain). The information displayed is the average ± standard deviation of three separate tests. * *p* < 0.05, ** *p* < 0.01, compared to the 10:0 ratio group using Dunnett’s test and ANOVA.

**Figure 4 ijms-26-01534-f004:**
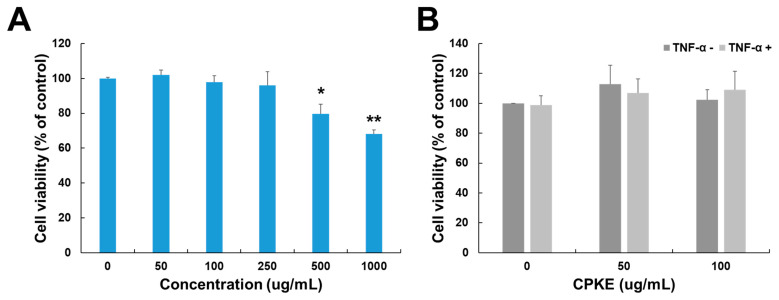
Evaluation of the viability of HaCaT cells subjected to varying CPKE treatment concentrations. (**A**) Cells were treated with CPKE treatment at specified concentrations for 24 h. (**B**) The cells were pretreated with CPKE at doses of 0, 50, and 100 µg/mL for 30 min. Ten minutes later, cells were stimulated with 10 ng/mL of TNF-α. Following that, the MTT test was used to evaluate cell viability. The mean ± SD of three separate experiments is displayed. Statistically significant difference from the control group, * *p* < 0.05, ** *p* < 0.01, compared to each group using Dunnett’s test and ANOVA.

**Figure 5 ijms-26-01534-f005:**
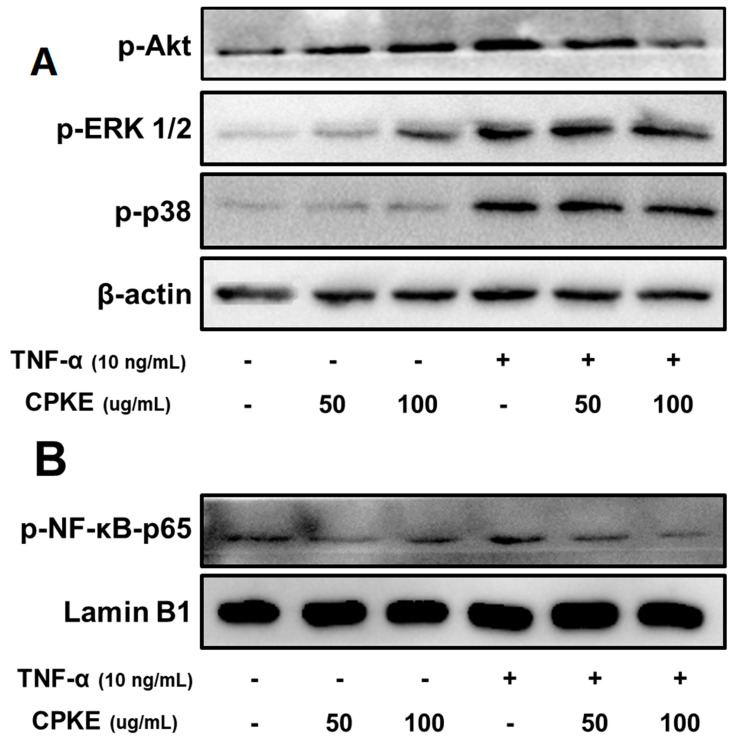
Suppression of mitogen-activated protein kinases (MAPK) and nuclear factor-kappa B by CPKE. For 30 min, HaCaT cells were pre-treated with CPKE at doses of 0, 50, and 100 µg/mL. This was followed by 30 min of stimulation with 10 ng/mL of TNF-α. (**A**) MAPKs like ERK1/2 and p38 were assessed by Western blot analysis. The inhibitory dosage was linked with the inhibition of Akt degradation after TNF-α stimulation of HaCaT cells. (**B**) The inhibition of the NF-κB p65 pathway in TNF-α induced HaCaT cells was assessed by Western blot analysis. The information displayed is the average ± standard deviation of three separate tests.

**Figure 6 ijms-26-01534-f006:**
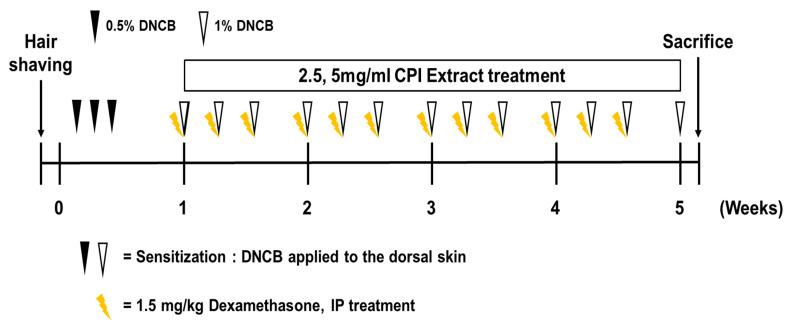
Experimental schedule for the induction of AD. Schedule of experiments for AD induction. Five groups of mice were created, with seven mice in each group. Each group is as follows: Vehicle-treated (sham) group 1; vehicle-treated (control) group 2; 2.5 mg/mL and 5 mg/mL CPKE with DNCB, respectively; and dexamethasone with DNCB (positive control) group 5 are the groups. DNCB was administered to the dorsal skin and ears to cause immunological and cutaneous lesions resembling AD.

**Figure 7 ijms-26-01534-f007:**
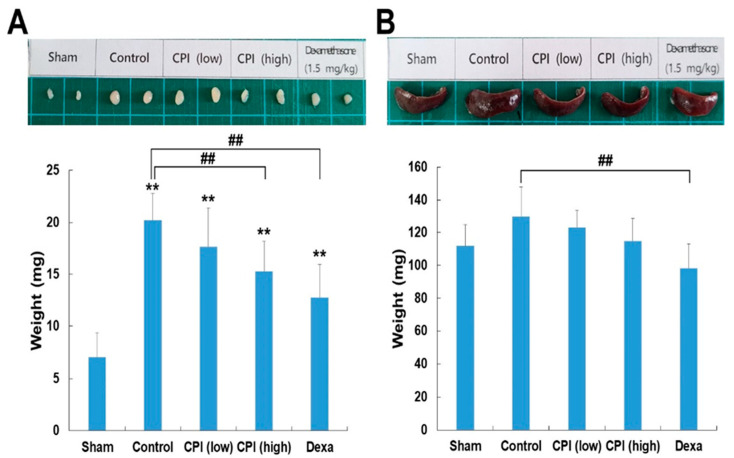
Inhibitory effects of CPKE on DNCB-induced AD skin symptoms in BALB/c mice. Vehicle (sham), DNCB + vehicle (control), DNCB + CPKE (low), DNCB + CPKE (high), and positive control (DEX) were the five groups into which the mice were divided. The sizes of (**A**) lymph nodes and (**B**) spleen organs were compared using photographs. Organ and whole-body weights were measured for each group, with a total of seven mice analyzed within each group. The mean ± SD of three separate experiments is displayed. Statistically significant difference from the control group, ** *p* < 0.01, compared to Sharm group, and ^##^ *p* < 0.05, compared to Control group using Dunnett’s test and ANOVA.

**Figure 8 ijms-26-01534-f008:**
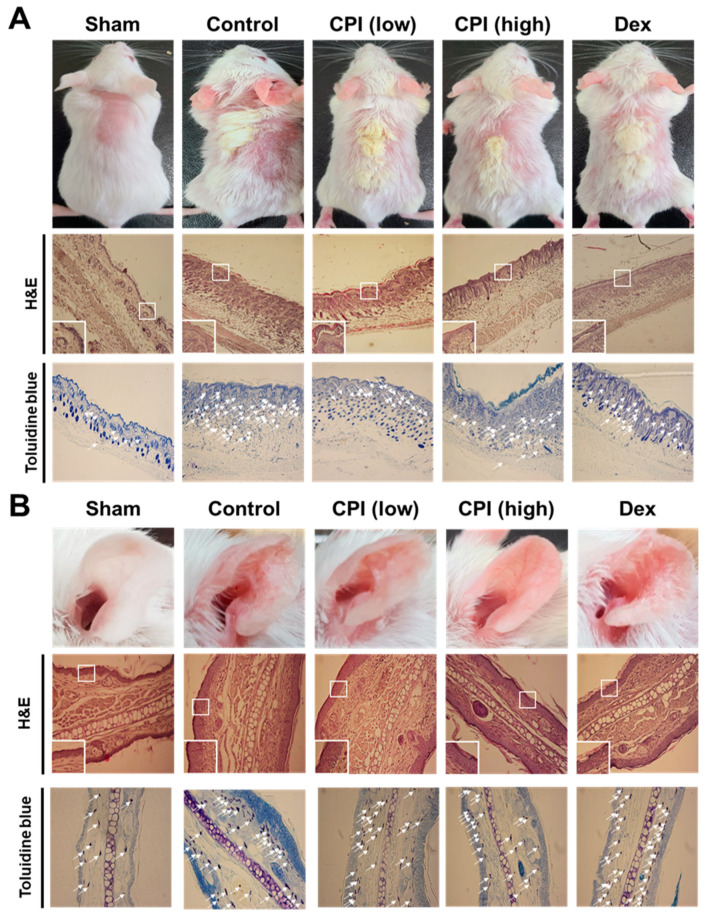
Impact of CPKE on histological alterations in back and ear lesions. (**A**) Back and (**B**) ear lesions were excised and preserved in a solution of 10% formaldehyde for fixation. Skin specimens were sectioned and subjected to staining with hematoxylin and eosin, as well as toluidine blue. Arrows indicate the immunological cells. Images were captured using a standard light microscope with a magnification of ×100 for the skin and ×200 for the ears.

## Data Availability

Not applicable.

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
