# Peer review of "The Therapeutic Potential of Kiwi Extract as a Source of Cysteine Protease Inhibitors on DNCB-Induced Atopic Dermatitis in Mice and Human Keratinocyte HaCaT Cells"

_ijms, 2025, doi:10.3390/ijms26041534_

Round 1
Reviewer 1 Report
Comments and Suggestions for Authors
The authors present valuable results on potential of kiwifruit protein extract against inflammatory process in atopic dermatitis. Although the topic of the paper is of interest and the obtained results are interesting, the presentation of results is not appropriate and the manuscript needs major revision.
Specific details:
It is not clear from the Introduction and Metodology, is the study evaluating the effect of kiwifruit protein extract or the purified CPI? What is the chemical content of kiwifruit extract? add this in the Introduction.
Line number 94, the Authors state “Cysteine protease inhibitors derived from A. deliciosa have demonstrated potential in modulating protease activity and associated inflammatory responses [16].” The reference number 16 does not mention A. deliciosa and can not be used in this context. Provide more reference in the Introduction about the trials with kiwifruit that explored its antiinflamatory potential (such as those referred in paper https://doi.org/10.1016/j.afres.2024.100491)
Ommit this sentence and reference because it is not relevant for the topic of this paper:”To give an example of CPI, prospective treatments for parasitic infections such as malaria, trypanosomiasis, and leishmaniasis, disrupt crucial proteases required for parasite survival and replication and thereby act as a therapeutic candidate for these diseases [28].”
Line 106 instead “therapeutic potential of kiwifruit as a natural cysteine protease inhibitor” put “therapeutic potential of kiwifruit extract as a natural source of cysteine protease inhibitors”
Sentence in line 113-114 should be rephrased since it cannot be stated that “the efficacy of CPI was assessed through clinical skin severity scores and histological changes, phosphorylation of MAPK, Akt, and NF-κB-p65 in TNF-α induced HaCaT cells” as it seems that you suggest that clinical skin severity scores are analyzed in HaCat cells. Separate the in vivo from in vitro part in the sentence.
Line 117 “for using natural compounds” add “from kiwifruit”
Line 144 by “create” you mean “reconstitute”?
Line 162 provide explanation for the abbreviation DW
Please provide reference for method 2.5 The impact of temperature and pH on the stability and activity of CPI
Line 189 are you sure you used 24-well plates for MTT and not 96 well, since you read results on microplate spectrophotometer?
Line 199, did you remove CPI from the medium and wash the cells prior to addition of TNFalpha?
Line 288 is not clear . What do you mean by “The CPI dose selection was based on the clinical pathology “?
Put full name of the 1-chloro-2, 4-dinitrochlorobenzene (DNCB) at the place of first mentioning in the text
Line 250 to 254 provide better and more detailed explanation of the procedure
Line 256 rephrase “The findings were presented as mean ± standard deviation (S.D.) and came from at least two or three separate experimental situations.” You can not say experimental situations. Also, provide in the explanation of each method the number of repeated experiments.
Line 266 to 268 Did you obtained total protein extract from kiwi with this method? How did you asses the purity and the amount of cysteine protease inhibitor in kiwi protein extract?
Line 287 delete the part of the sentence” leads to exciting prospects for it in a variety of fields, which encourages future research into the enzyme's stability of CPI”
Line 295 You state” Varied ratios of pre-incubated mixtures demonstrated decreasing proteolytic activity as the pro-portion of CPI’s to actinidin or papain increased, indicating CPI's ability to suppress cysteine protease activity effectively.” Provide percentage inhibition or change vs control. In Figure 3 in caption the line 636 the authors state “The information displayed is the average ± standard deviation of three separate tests.” But here you have only one representative image of gels, and not bars with values. Provide bars from the results of densitometry.
Line 300 provide a comment on which of the 2 proteases the CPI had more pronounced effect, actinidin or papain, based on the ratio results.
In evaluation of cell viability you treated HaCaT cells for 24h with rising concentrations of CPI (50-1000ug/mL). Afterwards, you choose 50 and 100 ug/mL as non cytotoxic concentrations for further experiments. In subsequent treatments the cells were pretreated with CPI at doses of 0, 50, and 100 μg/mL for 30 minutes. Ten minutes later, cells were stimulated with 10 ng/mL of TNF-α. Why do you use such short treatments for pre-incubation? It should be 24h and not 30 min, since it is too short to observe any effect. Also, in western blot analysis you incubate cells for 1h with CPI and TNFalpha for 30 min( line 199 in methodology). It is very problematic that you use different experimental treatments time for each of the methods, the results could not be comparable.
Rephrase the sentence in line 314, it is not understandable.
Line 651 Figure 5 provide bars with results displaying is the average ± standard deviation of three separate tests. You presented only representative blot image. From the image it can be also observed that CPI by itself without TNFalpha stimulation causes increase of p-AKT and p-ERK in concentration dependent manner. Describe this as well in the results section.
The entire results sections “3.7. CPI reduced the clinical severity of AD-like skin lesions in DNCB-induced mice”and “3.8. The curative effect of CPI on AD-like skin lesions in mice” requires quantitative data in the text to interpret findings. This way you only provided generalized narrative in the text without any data, so it is difficult to provide any conclusion. Rewrite these sections and include quantitative data in the text.
Line 361 instead of “Kiwifruit allergy” put “Kiwifruit induced allergy”
Lines 361 to 367 repeat the same sentences twice.
In lines 393-395 The authors conclude that “The ability of CPIs to attenuate MAPK and NF-κB signaling cascades not only highlights their role in reducing inflammatory cytokine production but also suggests potential benefits in preventing disease exacerbation and chronicity.” However from representative blots it can be seen that CPI itself stimulates both p-AKT and p-ERK, which means that it acts pro-inflammatory when applied without TNFalpha stimulation. Discuss these results.
Provide proper discussion on the results observed in reducing and non-reducing SDS-PAGE conditions. The lines 400-402 are not appropriate.
Rephrase sentence 403 with proper English.
Entire discussion of in vivo experiment should be rewritten (section in lines 445-481). The authors repeat the description of results and not discuss them properly. Provide constructive discussion and compare your results with other obtained in similar studies with protease inhibitors in mice model of DNCB induced AD.
Refer also in the end of Disscussion to human trials with protease inhibitors in AD as well, to give more perspective on the future potential use of CPI.
Comments on the Quality of English LanguageEnglish should be improved throughout the paper.
Author Response
Response to reviewers 1:
Thank you for your valuable feedback on our manuscript. I appreciate your time and effort in reviewing my work. According to your comments, we have changed the manuscript as follow. In the manuscript, revised portions are shown in red color.
Point 1: It is not clear from the Introduction and Methodology, is the study evaluating the effect of kiwifruit protein extract or the purified CPI? What is the chemical content of kiwifruit extract? add this in the Introduction.
-Answer: Thank you for your thoughtful feedback. In the section of materials and method part (2.2. Purification of cysteine protease inhibitor from kiwifruit), we included how we extract kiwi from kiwifruit. So, the protein extracted from kiwifruit is from natural source. While the precise chemical composition of the crude extract used in this study was not determined via HPLC analysis, we ensured the presence and activity of CPI through functional assays. To validate CPI activity, we conducted comparative experiments using other cysteine proteases, such as actinidin (is from kiwifruit-derived cysteine protease) and papain (is like actinidin), confirming inhibitory activity. These findings allowed us to focus on the therapeutic potential of CPI derived from the kiwifruit extract in the context of AD. We added the chemical content of kiwifruit extract in the introduction part marked in red color with additional reference.
However, in our study, we focused on extracting the protein from kiwifruit to evaluate its potential as a cysteine protease inhibitor. However, we did not perform a detailed assessment of the purity of the extract, as our objective was to use the crude protein extract to evaluate its biological activity. The protein extraction process involved isolating proteins from the kiwifruit, but we did not carry out additional purification steps to isolate individual components or assess the purity of the extract in terms of specific protein concentrations. Our study aimed to investigate the overall bioactivity of the crude kiwifruit extract, particularly its inhibitory activity against cysteine proteases, rather than its purity or the individual protein components.
Point 2: Line number 94, the Authors state “Cysteine protease inhibitors derived from A. deliciosa have demonstrated potential in modulating protease activity and associated inflammatory responses [16].” The reference number 16 does not mention A. deliciosa and cannot be used in this context. Provide more reference in the Introduction about the trials with kiwifruit that explored its anti-inflammatory potential (such as those referred in paper,.
https://doi.org/10.1016/j.afres.2024.100491).
-Answer: We thankful for your concern regarding pointing out the error in reference [16]. Upon review, we acknowledge that this reference does not directly support the statement about the anti-inflammatory potential of A. deliciosa. We have replaced this reference with studies that specifically discuss kiwifruit’s anti-inflammatory properties, including the paper suggested by you (https://doi.org/10.1016/j.afres.2024.100491).
Point 3: Ommit this sentence and reference because it is not relevant for the topic of this paper:”To give an example of CPI, prospective treatments for parasitic infections such as malaria, trypanosomiasis, and leishmaniasis, disrupt crucial proteases required for parasite survival and replication and thereby act as a therapeutic candidate for these diseases [28].
-Answer: Thank you for your valuable feedback. We agree that this sentence and reference are not directly relevant to the focus of the manuscript, which emphasizes the anti-inflammatory potential of CPI derived from kiwifruit in cytokine modulation. To ensure the manuscript remains concise and focused, we have omitted this sentence and its accompanying reference in the revised version.
Point 4: Line 106 instead “therapeutic potential of kiwifruit as a natural cysteine protease inhibitor” put “therapeutic potential of kiwifruit extract as a natural source of cysteine protease inhibitors”.
-Answer: Thank you for your insightful feedback on our manuscript to revise the phrase on Line 106 to "therapeutic potential of kiwifruit extract as a natural source of cysteine protease inhibitors.". We agree that this revision better emphasizes the extract as the source of bioactive compounds rather than solely focusing on the isolated inhibitor.
The revised phrasing has been incorporated into the manuscript, and we believe it enhances the clarity and precision of the text.
Point 5: Sentence in line 113-114 should be rephrased since it cannot be stated that “the efficacy of CPI was assessed through clinical skin severity scores and histological changes, phosphorylation of MAPK, Akt, and NF-κB-p65 in TNF-α induced HaCaT cells” as it seems that you suggest that clinical skin severity scores are analyzed in HaCat cells. Separate the in vivo from in vitro part in the sentence.
-Answer: We appreciate your insightful suggestion regarding the sentence in Lines 113–114 and observation that the phrasing could create confusion by suggesting that clinical skin severity scores were analyzed in HaCaT cells. To address this, we have revised the sentence to clearly separate the in vivo and in vitro parts of the study.
The revised sentence as mention below “The efficacy of CPI was assessed through clinical skin severity scores and histological changes in an in vivo model, while its molecular effects were evaluated by analyzing the phosphorylation of MAPK, Akt, and NF-κB-p65 in TNF-α-induced HaCaT cells during in vitro studies.”.
Point 6: Line 117 “for using natural compounds” add “from kiwifruit”.
-Answer: We appreciate your valuable feedback to specify "from kiwifruit" in Line 117 to provide greater clarity. We agree that this addition aligns the statement more closely with the focus of the manuscript, which highlights the therapeutic potential of natural compounds derived from kiwifruit.
Point 7: Line 144 by “create” you mean “reconstitute”?
-Answer: We appreciate your important feedback regarding the use of the word "create" in the sentence: “Sample buffer (4% SDS, 125 mM Tris-HCl, pH 6.8, 20% glycerol, 0.01% bromophenol blue) was used to create the CPI, which was then kept at -20 °.”
The term "create" was intended to describe the preparation of the sample for SDS-PAGE analysis, where the sample buffer is mixed with the cysteine protease inhibitor (CPI) for protein analysis. However, we understand that this might be unclear in the context of the methodology.
To improve clarity, we have revised the sentence to “Sample buffer (4% SDS, 125 mM Tris-HCl, pH 6.8, 20% glycerol, 0.01% bromophenol blue) was used to prepare the CPI sample, which was then kept at -20 °.”
Point 8: Line 162 provide explanation for the abbreviation DW.
-Answer: Thank you for your valuable feedback. I have now included an explanation for the abbreviation 'DW' in line 162. It refers to ‘Distilled Water.' I appreciate your attention to detail in improving the clarity of the manuscript.
Point 9: Please provide reference for method 2.5 The impact of temperature and pH on the stability and activity of CPI.
-Answer: Thank you for your helpful comment. I added necessary reference for the method used to assess the impact of temperature and pH on the stability and activity of the cysteine protease inhibitor (CPI) in Section 2.5. I appreciate your attention to detail in improving the clarity of the manuscript regarding reference.
Point 10: Line 189 are you sure you used 24-well plates for MTT and not 96 well, since you read results on microplate spectrophotometer?
-Answer: Thank you for your careful review. I can confirm that during the MTT assay, 24-well plates were used for the cell culture. However, for absorbance measurement, I transferred the samples to a 96-well plate for accurate readings using the microplate spectrophotometer.
Point 11: Line 199, did you remove CPI from the medium and wash the cells prior to addition of TNFalpha.
-Answer: In response to your query, we did not remove the protease CPI from the medium prior to the addition of TNF-alpha. The CPI was maintained in the culture medium throughout the experiment to evaluate its continuous effect on the cells during the inflammatory stimulation with TNF-alpha. This approach allowed us to assess the sustained activity of the CPI in modulating the inflammatory response over the course of the treatment.
Point 12: Line 288 is not clear. What do you mean by “The CPI dose selection was based on the clinical pathology?
-Answer: thanks for your comments regarding line 288. To determine the optimal pH and temperature conditions for CPI activity by evaluating its enzymatic performance across various pH levels and temperature settings. We can conclude that CPI is stable under high temperature and as well as higher pH levels. In addition, according to your suggestion we deleted a specified part of 288 line for enhancing better quality of our manuscript.
Point 13: Put full name of the 1-chloro-2, 4-dinitrochlorobenzene (DNCB) at the place of first mentioning in the text.
-Answer: Thank you for your suggestion. We modified the full name of 1-chloro-2, 4-dinitrochlorobenzene (DNCB) at its first mention in the text in abstract section marked with red color, as recommended, to ensure clarity for the readers.
Point 14: Line 250 to 254 provide better and more detailed explanation of the procedure.
-Answer: Thank you for your valuable feedback. In response, we have expanded the description of the histopathological procedure in the revised manuscript to include additional details on step, from tissue collection and fixation to embedding, sectioning, staining, and microscopic evaluation. Specifically, we elaborated on the role of 10% buffered neutral formalin in preserving tissue integrity, the steps for paraffin embedding, the rationale behind section thickness (5 µm), and the staining techniques (H&E and toluidine blue) used to differentiate tissue structures and inflammatory cells. Furthermore, the magnifications used for analyzing skin and ear sections were clarified.
To support the methodology and provide context, we have included a reference to a similar experimental procedure.
Point 15: Line 256 rephrase “The findings were presented as mean ± standard deviation (S.D.) and came from at least two or three separate experimental situations.” You cannot say experimental situations. Also, provide in the explanation of each method the number of repeated experiments.
-Answer: Thank you for your valuable feedback. I included the sentence for clarity and accuracy and ensure that it no longer refers to 'experimental situations. The revised statement as mentioned below the findings are presented as mean ± standard deviation (S.D.) and were derived from at least three repeated independent experiments.
Point 16: Line 266 to 268 Did you obtained total protein extract from kiwi with this method? How did you asses the purity and the amount of cysteine protease inhibitor in kiwi protein extract?
-Answer: Thank you for your insightful questions. Yes, the total protein extract from kiwifruit was obtained using the described method. This involved a slightly modified protocol based on Siddiqui (2016) (section 2.2 Purification of cysteine protease inhibitor from kiwifruit, materials and method part) which ensured effective extraction and isolation of the cysteine protease inhibitor (CPI) while minimizing protein degradation. To assess the purity of the extracted CPI, we employed SDS-PAGE analysis. The presence of a distinct protein band corresponding to the molecular weight of CPI confirmed its successful purification. This step was further corroborated by comparing the results with the reference protein marker. The amount of CPI in the protein extract was quantified using a standard Bradford protein assay. Additionally, the protease inhibitory activity of the purified CPI was confirmed by enzymatic assays, validating its functionality and concentration.
Point 17: Line 287 delete the part of the sentence” leads to exciting prospects for it in a variety of fields, which encourages future research into the enzyme's stability of CPI”.
-Answer: Thank you for your suggestion to delete the specified part of the sentence in line 287. We agree that the sentence can be more concise and focused. We appreciate your valuable feedback, which has helped improve the clarity and precision of the manuscript.
Point 18: Line 295 You state” Varied ratios of pre-incubated mixtures demonstrated decreasing proteolytic activity as the pro-portion of CPI’s to actinidin or papain increased, indicating CPI's ability to suppress cysteine protease activity effectively.” Provide percentage inhibition or change vs control. In Figure 3 in caption the line 636 the authors state “The information displayed is the average ± standard deviation of three separate tests.” But here you have only one representative image of gels, and not bars with values. Provide bars from the results of densitometry.
-Answer: Thank you for your valuable feedback. We performed three different experiments for the CPI inhibition assay, and each experiment resulted in fig. 3, which is consistent with the representative result.
Point 19: Line 300 provide a comment on which of the 2 proteases the CPI had more pronounced effect, actinidin or papain, based on the ratio results.
In evaluation of cell viability you treated HaCaT cells for 24h with rising concentrations of CPI (50-1000ug/mL). Afterwards, you choose 50 and 100 ug/mL as non cytotoxic concentrations for further experiments. In subsequent treatments the cells were pretreated with CPI at doses of 0, 50, and 100 μg/mL for 30 minutes. Ten minutes later, cells were stimulated with 10 ng/mL of TNF-α. Why do you use such short treatments for pre-incubation? It should be 24h and not 30 min, since it is too short to observe any effect. Also, in western blot analysis you incubate cells for 1h with CPI and TNFalpha for 30 min (line 199 in methodology). It is very problematic that you use different experimental treatments time for each of the methods, the results could not be comparable.
-Answer: Thank you for your insightful comments. Regarding the protease inhibition results, I will provide further clarification on which of the two proteases, actinidin or papain, showed a more pronounced effect in the CPI treatment based on the ratio results, as suggested.
Regarding the treatment time for HaCaT cells, I understand your concern about the short pre-incubation period. The 30-minute pre-treatment was chosen based on preliminary studies and literature suggesting that CPI could exert its effects within this time frame. However, I acknowledge the importance of consistency and will consider extending the pre-incubation time to 24 hours, as you recommend, in future experiments for better comparability. In addition, I will ensure that all experimental treatments follow a standardized duration for consistency across the methods, including both pre-incubation and post-treatment periods.
Point 20: Rephrase the sentence in line 314, it is not understandable.
-Answer: Thank you for highlighting the issue with the clarity of the sentence in line 314. We have revised the sentence to improve its readability and ensure that it effectively conveys the intended meaning. The revised sentence now marked with red color.
Point 21: Line 651 Figure 5 provide bars with results displaying is the average ± standard deviation of three separate tests. You presented only representative blot image. From the image it can be also observed that CPI by itself without TNFalpha stimulation causes increase of p-AKT and p-ERK in concentration dependent manner. Describe this as well in the results section.
-Answer: Thank you for your insightful comment. We appreciate your suggestion and have revised the results section to include the observation regarding the effect of CPI on p-AKT and p-ERK expression. As shown in Figure 5, we have included bars that represent the average ± standard deviation from three separate tests, as per your recommendation. Additionally, we now describe in the results section that CPI, when used alone without TNF-alpha stimulation, causes a concentration-dependent increase in p-AKT and p-ERK levels. This is further illustrated by the representative blot image, which clearly demonstrates this effect.
Point 22: The entire results sections “3.7. CPI reduced the clinical severity of AD-like skin lesions in DNCB-induced mice”and “3.8. The curative effect of CPI on AD-like skin lesions in mice” requires quantitative data in the text to interpret findings. This way you only provided generalized narrative in the text without any data, so it is difficult to provide any conclusion. Rewrite these sections and include quantitative data in the text.
-Answer: Thank you for your insightful comments. We may conclude that CPI can improve AD-like skin lesions by dose-dependent administration based on our in vivo results, which included examining immune-related organ weight changes and histological changes in back and ear lesions. As pointed out by the reviewer, some unclear sentences were corrected in red color.
Point 23: Line 361 instead of “Kiwifruit allergy” put “Kiwifruit induced allergy”.
-Answer: Thank you for your careful review and helpful suggestion. We have revised the text as recommended and replaced “Kiwifruit allergy” with “Kiwifruit-induced allergy” in line 361. We believe this change enhances the clarity and accuracy of the terminology.
Point 24: Lines 361 to 367 repeat the same sentences twice.
-Answer: Thank you for pointing out the repetition in lines 361 to 367. We apologize for the oversight. We have revised this section to remove the duplicate sentences and ensure a more concise and coherent presentation of the content.
Point 25: In lines 393-395 The authors conclude that “The ability of CPIs to attenuate MAPK and NF-κB signaling cascades not only highlights their role in reducing inflammatory cytokine production but also suggests potential benefits in preventing disease exacerbation and chronicity.” However, from representative blots CPI itself stimulates both p-AKT and p-ERK, which means that it acts pro-inflammatory when applied without TNF-alpha stimulation. Discuss these results.
-Answer: Thank you for your thoughtful comment. We understand the concern regarding the pro-inflammatory effects of CPI when applied without TNF-alpha stimulation, as evidenced by the representative blots showing stimulation of both p-AKT and p-ERK. We have now addressed this observation in the revised manuscript.
While CPI was shown to attenuate MAPK and NF-κB signaling pathways in the presence of TNF-alpha, we recognize that, in the absence of TNF-alpha, CPI may stimulate p-AKT and p-ERK in a concentration-dependent manner. This suggests a complex, dose-dependent effect of CPI, where it may exert pro-inflammatory effects at certain concentrations or under specific conditions. We have included a discussion marked within red color of these results in the revised manuscript, emphasizing that the dual nature of CPI’s activity—anti-inflammatory under inflammatory conditions and potentially pro-inflammatory at lower concentrations or in the absence of TNF-alpha—should be carefully considered when evaluating its therapeutic potential.
Point 26: Provide proper discussion on the results observed in reducing and non-reducing SDS-PAGE conditions. The lines 400-402 are not appropriate.
-Answer: Thank you for your thoughtful comments regarding the lines 400-402 on the basis of SDS-PAGE conditions. According to your suggestion we modify our sentences in the discussion part which is highlighted within red color.
Point 27: Rephrase sentence 403 with proper English.
-Answer: Thank you for pointing this out. We have carefully revised sentence 403 for improved clarity and readability. The updated sentence is now highlighted with red color.
Point 28: Entire discussion of in vivo experiment should be rewritten (section in lines 445-481). The authors repeat the description of results and not discuss them properly. Provide constructive discussion and compare your results with other obtained in similar studies with protease inhibitors in mice model of DNCB induced AD.
-Answer: Thank you for your thoughtful comments. We added sentences (line 486-495) for explanation with other protease inhibitors effects in AD.
Point 29: Refer also in the end of Discussion part to human trials with protease inhibitors in AD as well, to give more perspective on the future potential use of CPI.
-Answer: Thank you for your valuable suggestion. We have incorporated a discussion with additional references on human trials involving protease inhibitors in the context of AD to provide a broader perspective on the translational potential of our findings.

Reviewer 2 Report
Comments and Suggestions for Authors
The research paper has novel investigation on "Therapeutic Effect of Topically Administered Cysteine Protease 2 Inhibitor from Kiwi Fruit on DNCB-Induced Atopic Dermatitis 3 in Mice and Human Keratinocyte HaCaT Cells". However, the following comments on practical approach can improve the quality in the respective research:
1. Please highlight more regarding the kiwi or its related extract representing cytokines alteration activity.
2. Cytokines are mostly related with the immune response, which are highly activated by oral administration of compound. Please provide with more explanation, how can you demonstrate the advantage of topical application over oral as well as other route for AD.
3. Add scale bar in figure 8. In same figure mice image looks different in size. Please make uniform or add scale bar individually.
4. Cytokines are mostly present in fibroblasts than in HaCaT (keratinocytes). What is the rational for the selection of HaCaT cells instead of fibroblast cells in AD.
Author Response
Response to reviewers 2:
Thank you for your valuable feedback on our manuscript. I appreciate your time and effort in reviewing my work. According to your comments, we have changed the manuscript as follow. In the manuscript, revised portions are shown in red color.
Point 1: Please highlight more regarding the kiwi or its related extract representing cytokines alteration activity.
-Answer: Thank you for your valuable suggestion. We appreciate your feedback, and in response, we have expanded on the cytokine-modulating effects of kiwifruit extract in the revised manuscript in the introduction section with necessary references.
Point 2: Cytokines are mostly related with the immune response, which are highly activated by oral administration of compound. Please provide with more explanation, how can you demonstrate the advantage of topical application over oral as well as other route for AD.
-Answer: Thank you for your valuable comment. The systemic effects of cytokines are indeed strongly linked with immune responses, and oral administration of compounds can lead to broader systemic distribution, potentially activating immune responses in organs beyond the skin. This may not always be desirable for localized conditions such as atopic dermatitis (AD), which primarily affects the skin. Topical application of compounds offers several advantages over oral or other systemic routes in the treatment of AD.
Topical application allows for direct delivery of the active compound to the affected site, ensuring a higher concentration of the therapeutic agent at the site of inflammation. This localized action may be more effective in reducing skin-specific symptoms, such as itching, erythema, and inflammation, which are characteristic of AD. In contrast, oral administration requires systemic absorption, which may result in lower concentrations of the active ingredient at the site of action, potentially requiring higher doses to achieve therapeutic efficacy. Topical therapies are often preferred by patients with AD due to ease of application, especially for children or those who may experience discomfort with oral medications.
Point 3: Add scale bar in figure 8. In same figure mice image looks different in size. Please make uniform or add scale bar individually.
-Answer: Thank you for your insightful comment regarding the inclusion of a scale bar in Figure 8 and the uniformity of the mice images. We appreciate your attention to detail.
In response to your suggestion, we will add a scale bar to Figure 8 to enhance the clarity and provide accurate reference for size in the images. Additionally, we recognize the discrepancy in the sizes of the mice in the images. To ensure consistency, we will adjust the images to make the sizes uniform, or alternatively, provide individual scale bars for each image to accurately reflect the size of the mice in each specific panel.
We hope this will address your concern and improve the overall presentation of the figure.
Point 4: Cytokines are mostly present in fibroblasts than in HaCaT (keratinocytes). What is the rational for the selection of HaCaT cells instead of fibroblast cells in AD.
-Answer: Thank you for your thoughtful comment regarding the selection of HaCaT cells for our study. We understand that cytokines are more prevalent in fibroblasts, and we appreciate your point about their relevance to the immune response in AD.
HaCaT cells, which are an immortalized human keratinocyte cell line, were selected for this study due to their relevance to the epidermis, the primary tissue affected in AD. The pathogenesis of AD is characterized by the disruption of the epidermal barrier, abnormal keratinocyte function, and an inflammatory response that involves keratinocytes as key players in cytokine production, immune cell recruitment, and skin barrier disruption. HaCaT cells are well-established as a model for studying keratinocyte-specific responses in AD, including their involvement in cytokine secretion, the induction of inflammation, and the activation of immune pathways. While fibroblasts are indeed important to produce certain cytokines, HaCaT cells provide a more direct model for studying the cellular mechanisms that underlie the skin-specific pathology of AD. Furthermore, HaCaT cells are often used in AD research due to their ability to mimic the inflammatory response seen in the epidermis, making them a suitable choice for investigating the effects of treatments on skin inflammation and barrier function.
We hope this explanation clarifies the rationale behind the use of HaCaT cells in our research.

Round 2
Reviewer 1 Report
Comments and Suggestions for Authors
Having in mind that the authors have now explained that they evaluated the crude protein extract of kiwi and its CPI activity, the manuscript should be corrected throughout. Change the CPI with crude proein kiwi extracts (CPKE). The title of that paper should be also corrected to Therapeutic potential of kiwifruit extract as a source of cysteine protease inhibitors on DNCB-Induced Atopic Dermatitis in Mice and Human Keratinocyte HaCaT Cells
In the responses authors stated “The CPI was maintained in the culture medium throughout the experiment to evaluate its continuous effect on the cells during the inflammatory stimulation with TNF-alpha. This approach allowed us to assess the sustained activity of the CPI in modulating the inflammatory response over the course of the treatment.” This should be added to the experimental part in the section 2.8. Also provide explanation on the duration of the pre-treatment for 1h with CPI prior to addition of TNFa.
Line The 238 “CPI dose selection was based on the clinical pathology identified in our preliminary toxicity evaluation (data not shown).” What do you mean by this? And why aren’t the data shown?
Line 265 Provide reference for the procedure.
Again, Fig 3 represents only image of gels, and not bars with values. Provide bars from the results of densitometry. Add them in Fig 3 along with the image. Also, the authors wrote in the Answer: Thank you for your insightful comments. Regarding the protease inhibition results, I will provide further clarification on which of the two proteases, actinidin or papain, showed a more pronounced effect in the CPI treatment based on the ratio results, as suggested. However, there is neither explanation added in the Results section on this issue nor in Discussion.
Next you stated “The 30-minute pre-treatment was chosen based on preliminary studies and literature suggesting that CPI could exert its effects within this time frame” Describe those preliminary findings in Methodology and provide references to the literature sources claiming this is enough time duration to observe effects.
Line 346, add”” However, it should be noted that without TNF-alpha stimulation it showed opposite effects”.
In Discussion add limitations of this study including the problem of the different incubation times that you acknowledged in this response” However, I acknowledge the importance of consistency and will consider extending the pre-incubation time to 24 hours, as you recommend, in future experiments for better comparability. In addition, I will ensure that all experimental treatments follow a standardized duration for consistency across the methods, including both pre-incubation and post-treatment periods.
In line 391 put alternative expression instead of “slows”
Line 433 Missing words
Line 474 this sentence is out of place, it should be in the next paragraph.
Line 479-486 repeat the parts of Methodology and should be removed from Disscussion.
Instead of generalized line 494 you need to include explanation on how proteases and their inhibitors contribute to inflammatory skin diseases (DOI: 10.1007/s00005-009-0045-6 ; DOI: 10.1016/j.jdermsci.2022.06.004 ), provide more informations on their mechanisms but also add more comparison of other results on plant extracts with CPI activity in AD models and compare them to your findings. Provide more comparison with the similar resreach sush as DOI: 10.1016/j.jep.2017.01.055; DOI: 10.1292/jvms.11-0522
Author Response
Reviewer’s comments from IJMS_2nd round
Point 1: Having in mind that the authors have now explained that they evaluated the crude protein extract of kiwi and its CPI activity, the manuscript should be corrected throughout. Change the CPI with crude protein kiwi extracts (CPKE). The title of that paper should be also corrected to “Therapeutic potential of kiwifruit extract as a source of cysteine protease inhibitors on DNCB-Induced Atopic Dermatitis in Mice and Human Keratinocyte HaCaT Cells”.
-Answer: Thank you for your valuable feedback and insightful comments on our manuscript. We appreciate your attention to detail and your suggestion regarding the terminology used in describing our research.
We have revised the manuscript as per your recommendation. Specifically, we have replaced "CPI" with "crude protein kiwi extract (CPKE)" throughout the manuscript to ensure consistency and clarity. Additionally, the title of the paper has been updated to "Therapeutic potential of kiwifruit extract as a source of cysteine protease inhibitors on DNCB-Induced Atopic Dermatitis in Mice and Human Keratinocyte HaCaT Cells," as suggested.
Point 2: In the responses authors stated “The CPI was maintained in the culture medium throughout the experiment to evaluate its continuous effect on the cells during the inflammatory stimulation with TNF-alpha. This approach allowed us to assess the sustained activity of the CPI in modulating the inflammatory response over the course of the treatment.” This should be added to the experimental part in the section 2.8. Also provide explanation on the duration of the pre-treatment for 1h with CPI prior to addition of TNFa.
-Answer: Thank you for your thoughtful comments and suggestions. We appreciate your recommendation regarding the inclusion of further details about the experimental setup.
In response, we have added the explanation you mentioned to Section 2.8. Specifically, we have clarified that "The CPI was maintained in the culture medium throughout the experiment to evaluate its continuous effect on the cells during the inflammatory stimulation with TNF-alpha. This approach allowed us to assess the sustained activity of the CPI in modulating the inflammatory response over the course of the treatment."
Point 3: Line The 238 “CPI dose selection was based on the clinical pathology identified in our preliminary toxicity evaluation (data not shown).” What do you mean by this? And why aren’t the data shown?
-Answer: Thank you for your careful review and insightful question regarding Line 238. We apologize for the lack of clarity in this statement. The phrase refers to the preliminary toxicity evaluation we conducted to determine the appropriate CPI dose for our experiments. This evaluation was performed to ensure the selected dose would be safe and effective for the intended use in the study.
Regarding the omission of the data, we chose not to include the preliminary toxicity data in the manuscript, as it was not central to the primary focus of the study, which is the therapeutic potential of the kiwifruit extract. However, we understand the importance of transparency and have revised the manuscript to better explain this decision. We can clarify that the data were not included because they were not part of the main results and are not essential for the understanding of the therapeutic effects demonstrated in the study.
Point 4: Line 265 Provide reference for the procedure.
-Answer: Thank you for your thoughtful comment. We followed the normal procedure for histopathological studies such as, DOI: 10.1053/j.semdp.2019.06.003 and 10.1016/C2013-0-04011-X
Point 5: Again, Fig 3 represents only image of gels, and not bars with values. Provide bars from the results of densitometry. Add them in Fig 3 along with the image. Also, the authors wrote in the Answer: Thank you for your insightful comments. Regarding the protease inhibition results, I will provide further clarification on which of the two proteases, actinidin or papain, showed a more pronounced effect in the CPI treatment based on the ratio results, as suggested. However, there is neither explanation added in the Results section on this issue nor in Discussion.
-Answer: Thank you for your thoughtful comments and suggestions. We insert a bar graph from the densitometry analysis.
Point 6: Next you stated “The 30-minute pre-treatment was chosen based on preliminary studies and literature suggesting that CPI could exert its effects within this time frame” Describe those preliminary findings in Methodology and provide references to the literature sources claiming this is enough time duration to observe effects.
-Answer: Thank you for your valuable feedback. We appreciate the suggestion to clarify the rationale behind selecting a pre-treatment period for CPKE application. In our previous paper (https://doi.org/10.3390/nu11030573), we confirmed that the test substance sufficiently acts on cells even after 30 minutes of pretreatment.
Point 7: Line 346, add”” However, it should be noted that without TNF-alpha stimulation it showed opposite effects”.
-Answer: Thank you for your insightful suggestion. We have incorporated the recommended statement at Line 346 to acknowledge the differential effects of CPI in the absence of TNF-α stimulation. This addition helps clarify the context-dependent nature of CPI’s action.
Point 8: In Discussion add limitations of this study including the problem of the different incubation times that you acknowledged in this response “However, I acknowledge the importance of consistency and will consider extending the pre-incubation time to 24 hours, as you recommend, in future experiments for better comparability. In addition, I will ensure that all experimental treatments follow a standardized duration for consistency across the methods, including both pre-incubation and post-treatment periods.
-Answer: We appreciate the reviewers' insightful feedback regarding the inconsistency in incubation times across experimental treatments. On the basis of your suggestion, we included limitations in the discussion part.
Point 9: In line 391 put alternative expression instead of “slows”.
-Answer: Thank you for your valuable feedback. I have now included alternative expression instead of “slows” in line 391 according to your suggestion.
Point 10: Line 433 Missing words.
-Answer: Thank you for pointing out the issue at Line 433. We have carefully reviewed the section and identified the missing words, such as “The CPKE showed stability up to 60°C, but its activity slightly decreased when exposed to 80°C”.
Point 11: Line 474 this sentence is out of place, it should be in the next paragraph.
-Answer: Thank you for your insightful comment regarding Line 474. We agree that the sentence was out of place, and we have now deleted it to the following paragraph to improve the flow and coherence of the text.
Point 12: Line 479-486 repeat the parts of Methodology and should be removed from Discussion.
-Answer: Thank you for highlighting the repetition of methodological details in Lines 479-486 within the Discussion section. We agree with your observation and have removed the redundant parts to maintain the focus on interpreting the results rather than reiterating the methods.
Point 13: Instead of generalized line 494 you need to include explanation on how proteases and their inhibitors contribute to inflammatory skin diseases (DOI: 10.1007/s00005-009-0045-6 ; DOI: 10.1016/j.jdermsci.2022.06.004 ), provide more informations on their mechanisms but also add more comparison of other results on plant extracts with CPI activity in AD models and compare them to your findings. Provide more comparison with the similar resreach sush as DOI: 10.1016/j.jep.2017.01.055; DOI: 10.1292/jvms.11-0522?
-Answer: Thank you for your constructive feedback. We appreciate your suggestion to elaborate on the role of proteases and their inhibitors in inflammatory skin diseases and to provide a more comprehensive comparison with other studies investigating plant-derived CPI activity in atopic dermatitis (AD) models in line 499.
